



# Real time updating of the flood frequency distribution through data assimilation

Cristina Aguilar[1], Alberto Montanari[2], María José Polo[3]

[1]Fluvial dynamics and hydrology research group, Andalusian Institute of Earth System Research, University of Granada, Granada, 18006, Spain
[2]Department DICAM, University of Bologna, Bologna, 40136, Italy
[3]Fluvial dynamics and hydrology research group, Andalusian Institute of Earth System Research, University of Córdoba, Cordoba, 14071, Spain

*Correspondence to*: Cristina Aguilar (caguilar@ugr.es)

**Abstract**

We explore the memory properties of catchments for predicting the likelihood of floods basing on observations of average flows in pre-flood seasons. Our approach assumes that flood formation is driven by the superimposition of short and long term perturbations. The former is given by the short term meteorological forcing leading to infiltration and/or saturation excess, while the latter is originated by higher-than-usual storage in the catchment. To exploit the above sensitivity to long term perturbations a Meta-Gaussian model is implemented for updating a season in advance the flood frequency distribution, through a data assimilation approach. Accordingly, the peak flow in the flood season is predicted by exploiting its dependence on the average flow in the antecedent seasons. We focus on the Po River at Pontelagoscuro and the Danube river at Bratislava. We found that the shape of the flood frequency distribution is significantly impacted by higher-than-usual flows occurred up to several months earlier. The proposed technique may allow one to reduce the uncertainty associated to the estimation of flood frequency.

## 1 Introduction

The physical, chemical and ecological state of processes leading to the formation of river flow is characterised by persistence at several different time scales (Koutsoyiannis, 2014). In fact, anomalous conditions for such processes, such as those generated by extreme meteorological events, may produce





a long lasting impact on the river flow, depending on climatic and catchment behaviours (Lo and Famiglietti, 2010). For instance, flood generation is impacted by the initial soil moisture condition of the catchment, which may be in turn impacted by groundwater levels that are related to global catchment storage (Massari et al., 2014). Persistence can be exploited to improve river flow forecasting

at seasonal-to-interannual time scale. Furthermore, persistence provides useful indications to better understand the functioning of a catchment and the dynamics of the water cycle.

Indeed, the study of persistence is one of the most classical research endeavours in hydrology, since the early works by Rippl (1883) and Hazen (1914) on the estimation of the optimal storage for reservoirs. Hurst (1951) investigated the Nile River flows while working at the design of the Aswan Dam and

postulated that geophysical records may be affected by a complex form of persistence that may last for long time (O' Connell et al., 2015). Later on, Thomas and Fiering (1962) and Yevjevich (1963) introduced autoregressive models for annual and seasonal streamflow simulation therefore stimulating the development of subsequent models of increasing complexity for simulating hydrological persistence.

Recently, the attention has been focused on long term persistence (LTP), which is associated to the Hurst-Kolmogorov behaviour (Koutsoyiannis, 2011). LTP manifests itself through a power-law decay of the autocorrelation function of the process, which implies that the summation of the autocorrelation coefficients diverges to infinity (Montanari et al., 1997). LTP implies the possible presence of long term cycles (Beran, 1994), which in turn means that perturbations of hydrological processes may last for a

long time, therefore providing a possible explanation for the occurrence of clusters of extreme hydrological events, such as floods and droughts (Montanari, 2012). LTP also has implications in the study of climate change, as it is connected with an enhanced natural variability of climatic processes (Koutsoyiannis and Montanari, 2007).

While LTP has been long studied, limited attempts have been made to exploit LTP in data assimilation

procedures for improving streamflow forecasting. The motivation probably is that LTP is recognized to exert a significant impact on the river flow volume over long time scales, while its effect on the magnitude of single events is deemed to be negligible.





The present contribution aims to enhance our understanding of the persistence properties of river flows and to inspect the opportunity of exploiting persistence to improve seasonal river flow forecasting. By taking inspiration from the idea that the probability of extreme floods may be increased by long term stress, like higher than usual rainfall lasting for several months, the research question that we address

here can be stated as follows: Can higher than usual river discharges in the previous season be associated to a higher probability of floods in the subsequent high flow season? (Aguilar et al., 2016). The quantification of the effect of variable antecedent flows for different time lags on the occurrence of floods would help to assess how long a river remembers its past. From a technical point of view, we aim to propose a technique for updating a season in advance the flood frequency distribution estimated for a

given river, through a data assimilation approach, by exploiting the information provided by river flows in the pre-flood seasons.

It is interesting to highlight that the state of a catchment, and in particular its storage, is affected by previous precipitation. Therefore, it would be reasonable to exploit the information provided by previous rainfall rather than previous flows for the sake of updating the flood frequency distribution.

However, areal rainfall estimation for catchments with large extension and complex orography is affected by large uncertainty (Moulin et al., 2009). Therefore, we utilize here flows during pre-flood seasons as a proxy for catchment storage instead of rainfall. While the above assumption may be reasonable, one should consider that it may not hold when the river flows are impacted by massive regulation.

**2 Study sites and data sources**

We focus our attention on two large basins, namely, the Po river basin at Pontelagoscuro (Italy) and the Danube river basin at Bratislava (Slovakia). The Po River is the longest river entirely flowing in the Italian Peninsula (Fig. 1) with a catchment area of about 71000 $km^2$ at the delta. The average annual precipitation in the catchment is 78 $km^3$ in volume, of which 60% reaches the closure river cross-section

at Pontelagoscuro. The hydrological behavior of the Po River is described in detail in recent studies (Zanchettin et al., 2008; Montanari, 2012; Zampieri et al., 2015). The discharge pattern at Pontelagoscuro presents a mean annual flow of about 1470 $m^3s^{-1}$ and shows a typical pluvial regime,



and thus a strong seasonality with two flood seasons in spring and autumn (Fig. 2). An intense exploitation of water resources for irrigation, hydro-power production, civil and industrial use is found in the catchment. Even though water resources management is currently sustainable on average, critical situations are experienced during drought periods (Montanari, 2012).

The upper Danube basin drains from the northern side of the Alps and the southern area of the central European Highlands into Bratislava in a 131331 km$^2$ catchment area where the mean annual flow is about 2053 m$^3$s$^{-1}$. The hydrological behavior of the upper Danube basin can be found in detail in the literature (Nester et al., 2011; Blöschl et al., 2013). The average annual precipitation in the catchment is 123 km$^3$ and the discharge pattern shows a typical alpine regime and thus a strong seasonality with one

flood season in the summer (Fig. 2).

Daily discharge and monthly precipitation and temperature data for the Po and Danube river basins were analyzed in this study. The observation periods as well as descriptive statistics of the different time series are shown in Table 1. Discharge time series at Pontelagoscuro for the Po River and Bratislava for the Danube River were provided, respectively, by the Regional Agency for Environmental Protection

(ARPA)— Emilia Romagna, Hydro-meteorological Office and by Global Runoff Data Center (GRDC, 2011). The series are not affected by missing values. They correspond to a time span of 90 and 107 years for Po and Danube, respectively.

The Po river is regulated by the presence of several dams as reservoirs for hydroelectricity production, which are mainly located in the Alpine region. Also, the outflow from the lakes Como, Garda, Iseo, Idro

and Maggiore is regulated (Zanchettin et al., 2008). These regulation practices do not significantly impact on the trend and the low-frequency variability of the peak flows, while they may affect the low flows at daily and sub-daily time scale (Zampieri et al., 2015). The upper part of the Danube has been ideal for building hydropower plants and up to 59 dams are found along the river's first 1000 km. As stated in the Danube River Basin Management Plan stretches in the very upper part of the river may

present significant altered flows. (Maps 7a, b, c in DRBM, 2009). The effect of regulation on peak flows in Slovakia is deemed to be negligible, while low and average flows may be significantly impacted.



Precipitation and temperature time series were calculated based on weather data sets obtained from the HISTALP project (Auer et al., 2007). Only weather stations where sufficiently long data sets are available were used (Table 1). The study period was conditioned by the availability of discharge data even though both meteorological variables were available for a longer historical period. For each study

site, catchment area average precipitation and temperature time series were constructed using Thiessen polygons.

## 3 Methodology

In order to address the research question outlined in Section 1, namely, to verify the opportunity of updating the flood frequency distribution a season in advance by exploiting the information provided by

the river flow in a given pre-flood season, we first estimate the Hurst exponent (H) for both time series, to verify whether the hypothesis of the presence of LTP is supported by data evidence. Then, we turn to the analysis of the statistical dependence between the peak flow in the flood season and the average flow during the previous season, to empirically check whether updating the flood frequency distribution produces useful results.

### 3.1 Estimation of long term persistence

Time series with long-term memory or persistence exhibit a power-law decay of the autocorrelation function (Beran, 1994), that is:

$$\rho(k) \sim c_k k^{2H-2} \quad k \to \infty \tag{1}$$

where $\rho(k)$ is the autocorrelation function of the process at lag k, $c_k$ is a constant and $H \in [0\ 1]$ is the

Hurst exponent or the intensity of the LTP (Montanari et al., 1997). For a stationary process H is constrained in the range [0.5,1]. A value equal to 0.5 means absence of LTP; the higher the H, the higher the intensity of LTP.

In this work H was estimated by using different heuristic methods. In detail, we applied the rescaled range (R/S) analysis, the aggregated variance method (climacogram; see Dimitriadis and Koutsyiannis,

2015), and the differenced variance method. An extended description of numerous methodologies to





assess the persistence properties of time series to provide support to the possible presence of the Hurst-Kolmogorov behavior can be found in Taqqu et al. (1995), Montanari et al. (1996, 1997, 2000) and Koutsoyiannis (2003).

A strong seasonal component in the different hydrological variables in both study time series has been
reported by the literature (e.g. Montanari, 2012; Szolgayova et al., 2014; Zampieri et al, 2015). It is well known that a strong seasonality often implies the presence of periodic deterministic components in the data that can introduce a bias in LTP estimation (Montanari et al., 1997, 2000). Also, the presence of slowly decaying or increasing trends may induce a bias as well. Thus, prior to long-term memory assessments, all time series were detrended and deseasonalized. For each time series, a 366-term (for
daily data) and 13-term (monthly data) moving average for a trend approximation was applied, followed by a stable seasonal filter for removing of the seasonal cycle (Brockwell et al., 2002).

### 3.2 Analysis of the peak flow dependence on average flows during pre-flood seasons

In order to analyze the stochastic connection between the average river flows in the antecedent seasons and the average and peak flow in the flood season, a bivariate probability distribution function was
fitted. In what follows, random variables and their outcomes are identified with bold and un-bold characters, respectively. The yearly variables analyzed in this study were:

-The monthly mean flow in the given pre-flood season (independent or explanatory variable), $\mathbf{Q_m}$.

-The peak flow in the flood season or annual maximum daily flow (dependent variable), $\mathbf{Q_p}$.

-The mean daily flow in the flood season (dependent variable), $\mathbf{Q_{mf}}$.

First, the time series $Q_m(t)$, $Q_p(t)$ and $Q_{mf}(t)$ with sample size n, where n is the number of years in the observation period, are extracted from the observed datasets. Then, the Normal Quantile Transform (NQT) is applied in order to make their marginal probability distributions Gaussian, therefore obtaining the normalized observations $NQ_m(t)$ and $NQ_p(t)$ and $NQ_{mf}(t)$. Numerous applications of the NQT in hydrological studies can be found in the literature (e.g. Moran, 1970; Hosking and Wallis, 1988;
Montanari and Brath, 2004; Montanari, 2005; Montanari and Grossi, 2008; Bogner et al., 2012; Aguilar et al., 2016).





The NQT involves the following steps when we take $\mathbf{Q_m}$ as an example: (1) Sorting the sample of $Q_m(t)$ from the smallest to the largest observation, $Q_{m1}$, ..., $Q_{mn}$; (2) estimating the cumulative frequency $FQ_{mi}$ by using the Weibull plotting position (Stedinger et al., 1993); (3) for each $FQ_{mi}$ the standard normal quantile $NQ_{mi}$ is computed as $NQ_{mi} = G^{-1}(FQ_{mi})$, with G denoting the standard normal distribution and

$G^{-1}$ its inverse, and associated with the corresponding $Q_{mi}$. Thus, a discrete mapping of $Q_{mi}$ to its transformed counterpart $NQ_{mi}$ is obtained. In order to apply the inverse of the NQT for any $NQ_{mi}$, linear interpolation is applied to connect the points of the discrete mapping previously obtained. Bogner et al. (2012) propose different parametric and non-parametric approaches for the extrapolation of extreme values. In this study the region beyond the maximum and the minimum available $NQ_{mi}$ values is

covered by linear extrapolation.

Finally, a meta-Gaussian model (Kelly and Krzysztofowicz, 1997; Montanari and Brath, 2004) is fitted between the random explanatory variable and each dependent variable in their canonical form in the Gaussian domain. In what follows, we specify the equations for the peak flow as the dependent variable. We assume: (1) stationarity and ergodicity of both $\mathbf{NQ_m}$ and $\mathbf{NQ_p}$; and (2) that the cross dependence

between both $\mathbf{NQ_m}$ and $\mathbf{NQ_p}$ can be represented by the normal linear equation:

$$NQ_p(t) = \rho(\mathbf{NQ_m}, \mathbf{NQ_p})NQ_m(t) + N\varepsilon(t) \qquad (7)$$

where $\rho(\mathbf{NQ_m}, \mathbf{NQ_p})$ is the Pearson's cross correlation coefficient between $\mathbf{NQ_m}$ and $\mathbf{NQ_p}$, and $N\varepsilon$ is an outcome of the stochastic process $\mathbf{N\Theta}$, which is independent, homoscedastic, stochastically independent of $\mathbf{NQ_m}$ and normally distributed with zero mean and variance $1 - \rho^2(\mathbf{NQ_m}, \mathbf{NQ_p})$. The parameters of the

bivariate probability distribution function are the mean ($\mu(\mathbf{NQ_m})=0$ and $\mu(\mathbf{NQ_p})=0$), the standard deviation ($\sigma(\mathbf{NQ_m})=1$ and $\sigma(\mathbf{NQ_p})=1$) of the normalized series, and the Pearson's cross correlation coefficient between both normalized series, $\rho(\mathbf{NQ_m}, \mathbf{NQ_p})$. In the presence of dependence between $\mathbf{NQ_m}$ and $\mathbf{NQ_p}$, the correlation coefficient will be significantly different from zero. The bivariate Gaussian distribution implies that, for an arbitrary (observed) $NQ_m(t)$, the probability distribution function of $\mathbf{NQ_p}$

is Gaussian, with parameters (Eq. 8 and 9):

$$\mu(\mathbf{NQ_p}) = \rho(\mathbf{NQ_m}, \mathbf{NQ_p}) \cdot NQ_m(t) \qquad (8)$$



$$\sigma(\mathbf{NQ_p}) = \left(1 - \rho^2\left(\mathbf{NQ_m}, \mathbf{NQ_p}\right)\right)^{0.5} \tag{9}$$

Then, by taking the inverse of the NQT one can infer the updated probability distribution of $\mathbf{Q_p}$ conditioned to the observed outcome $Q_m(t)$.

In order to verify the validity of the linear model (Eq. 7), a goodness of fit based on the behavior of the
residuals is applied. Following the graphical approach proposed by Cook and Weisberg (1994), the residual plot of $N\varepsilon(t)$ versus $\rho(\mathbf{NQ_m}, \mathbf{NQ_p}) \cdot NQ_m(t)$ should not show any systematic trend under the target model. Curve trends or fan shape trends indicate non-linear cross dependence and variability of the variance of $\mathbf{N\Theta}$, respectively (Montanari and Brath, 2004).

The same methodology was applied for the other dependent variable considered in this study, $\mathbf{Q_{mf}}$.
Therefore, once the parameters of each distribution are computed, the probability distribution function of both the peak flow and the mean flow in the flood season can be updated after observing the mean flow in the considered pre-flood season.

In order to infer the actual impact of the dependence between peak flows and mean flow in the flood season with the mean flow in the pre-flood seasons, the unconditioned flood frequency distribution and
the updated distributions inferred for several higher-than-average values of mean flow (e.g. 70%, 80% and 95% quantiles) in a given pre-flood season were compared. We assume that peak flows can be adequately modeled through the Extreme Value Type 1 (EV1) distribution and we present a comparison between the unconditioned peak flows frequency distribution and the updated peak flows frequency distributions.

**3.2.1 Identification of the flood season**

According to previous studies in the literature, directional statistics (Mardia, 1972) represents an effective method for identifying the timing of hydrological extreme events (e.g. Castellarin et al., 2001; Cunderlik and Burn, 2002; Baratti et al., 2012). Following Bayliss and Jones (1993), the date of occurrence of an event i (e.g. maximum annual daily flow) can be transformed into a directional statistic
by converting the Julian date of occurrence, $Jd_i$, into an angular measure, $\theta_i$, through Eq. (2):





$$\theta_i = Jd_i \left( \frac{2\pi}{365} \right).$$ (2)

Each date of occurrence can then be written in polar coordinates by means of a vector with a unit magnitude and the direction specified by Eq. (2). Therefore, the $x_p$ and $y_p$ coordinates of the mean of the sample of n dates of occurrence can be computed with Eq. (3):

$$x_p = \frac{1}{n}\sum_{i=1}^{n} \cos(\theta_i) \qquad y_p = \frac{1}{n}\sum_{i=1}^{n} \sin(\theta_i)$$ (3)

The direction, $\bar{\theta}$, and magnitude, r, of the mean in polar coordinates can then be obtained by Eq. (4) and Eq. (5) respectively. Equation (4) gives a measure of the mean timing of the event for the sample of dates, and can be converted back to a mean Julian date, MD, through Eq. (2). Equation (5) indicates the regularity or seasonality of the phenomenon. Values of r close to one imply a strong regularity in the

dates of occurrence of the event considered. In contrast, values of r close to zero indicate a great dispersion and thus, a great inter-annual variability in the dates of occurrence of the event throughout the year.

$$\bar{\theta} = \arctan\frac{y_p}{x_p}$$ (4)

$$r = \sqrt{x^2 + y^2}$$ (5)

Finally, the limits of the occurrence of the phenomenon can quantitatively be identified by adding and subtracting to $\bar{\theta}$, the standard deviation in radians, $\sigma$, given by Eq. (6):

$$\sigma = \sqrt{-2Ln(r)}$$ (6)

Directional statistics was applied to the following variables in order to identify the flood season in each study site: 1) annual maximum series of daily flows (AMD); 2) high flow events defined from

frequency analysis as those events when the daily discharge exceeds the 95$^{th}$ percentile, $Q_{95}$, for longer than 15 days.



## 4 Results and discussion

### 4.1 Long term persistence estimation

The application of the heuristic methods for LTP estimation to deseasonalized and detrended time series is displayed in Table 2. H values above 0.71 were obtained for the mean daily river flows in both rivers

and thus, all three heuristic methods detect the presence of significant LTP in the time series. The intensity of LTP seems to be more or less the same for monthly flow data. Similarly, H values in monthly temperature data above 0.64 and 0.61 in Po and Danube, respectively, suggest the presence of LTP in both records. In contrast, the estimated H values in the monthly rainfall datasets are not sensibly higher than 0.5.

In general, these results agree with previous outcomes of long term persistence studies for the daily discharge of the Po at Pontelagoscuro (Montanari, 2012) as well as with previous studies in the daily river flows in an upstream tributary of the Po (H=0.71-0.81) and in the monthly rainfall registered at certain weather stations within the watershed (Montanari et al., 1996; 1997). Also, H values of the same order of magnitude were found by Szolgayova et al. (2014) for the rainfall (H=0.43-0.50) and

temperature (H=0.65-0.72) monthly time series in the upper Danube watershed at Bratislava.

### 4.2 Meta-Gaussian model for updating the flood frequency distribution

#### 4.2.1 Flood season identification

Figure 3 shows the results of the directional statistics applied to the extreme events in both rivers. In the Po river we can see a very low regularity (r≈0.1) and high dispersion (4 months) in the annual

maximum daily flows (AMD in Fig. 3) due to their possible occurrence in any of the two high flow seasons, spring and autumn, as depicted in Fig. 2. The seasonality increases to r values close to 0.8 for high flow events that mostly take place in autumn as already reported in previous studies (Zanchettin et al., 2008; Montanari, 2012).

In the Danube we find a considerable regularity in high flow events (r≈0.8) but a certain decrease in the

annual maximum flows (with r values of 0.4). Nevertheless, the 2-month dispersion in the date of occurrence is lower than in the Po river and corresponds to the length of the high flow season reported




in Fig.2. In view of these results we set October-November and May-July as the main flood seasons in the Po and Danube respectively.

As pre-flood season, we consider a 1-month period, which is long enough in order to reduce the effect of river regulation. We first set the month preceding the flood season (i.e., September and April for Po and Danube, respectively) as pre-flood season. Then, we repeat the analysis by making reference to the previous months, with the expectation that the statistical dependence decreases as the pre-flood season is moved back in the past.

### 4.2.2 Estimation of the Meta-Gaussian model

Table 3 shows the cross correlation coefficients $\rho(NQ_m, NQ_p)$ and $\rho(NQ_m, NQ_{mf})$ between the normalized dependent variables ($NQ_p$ and $NQ_{mf}$ in both study sites) and the explanatory variable ($NQ_m$) at each study site. In detail, we assumed that $Q_m$ is given by the monthly mean flow in each of the 9 months preceding the floods season (from September to January in the Po river and from April to August in the antecedent year in the upper Danube). Table 2 shows that the correlation coefficient decreases as the considered pre-flood season moves backward, as we expected. Besides, we appreciate significantly higher coefficients with the mean flow in the flood season ($\rho(NQ_m, NQ_{mf})$), than with the annual maximum daily flows ($\rho(NQ_m, NQ_p)$) in both rivers. For example, a cross correlation coefficient of 0.24 was obtained between $NQ_p$ and $NQ_m$ in the Po when the pre-flood season considered is September, compared to 0.39 between $NQ_{mf}$ and the same explanatory variable, $NQ_m$. Moreover, a continuous decreasing cross correlation coefficient is found as we move further apart the flood season and negative correlation in the Po river appears from May-June backwards.

The only anomalous correlation is found when considering the $Q_m$ in March as the explanatory variable for both dependent variables in the Danube. This month corresponds with both the peak in the snowmelt annual cycle in the catchment (Zampieri et al., 2015) and the steepest rising slope in the hydrograph (Fig. 2). Therefore the use of monthly mean flow might not be representative given the high variability in the daily flows along this month and the complexity of the processes that are affecting the streamflow (complex contribution from subsurface flow or from the runoff generated from snowmelt/precipitation).





A goodness-of-fit test was carried out for the Meta-Gaussian model by using residuals plots (Montanari and Brath, 2004). Fig. 4 shows the residuals for a time span of 4-months backwards the flood season at each study site. The residuals look homoscedastic, therefore confirming the satisfactory fit.

### 4.2.3 Flood frequency distribution updating

In order to decipher the technical benefit that can be gained by updating the flood frequency distribution through the proposed data assimilation procedure, we assumed that above average river flows are observed in the month preceding the flood season and then applied the Meta-Gaussian model to estimate the updated probability distribution. In detail, we assume that, in average, monthly flow corresponding to the 70%, 80% and 95% quantile is observed in September for the Po River and April for the Danube River.

Fig. 5 and 6 show the unconditioned and updated probability density functions (pdf) of the normalized peak flow (i.e., the peak flow transformed to the canonical Gaussian distribution). We can appreciate that the higher the cross correlation value, the lower the variability in the distribution of the normalized dependent variable and the higher the mean value. For example, in the Po River for the occurrence of the 95[th] quantile value in the normalized mean flow in September, the pdf is centered around a mean vaue of 0.4 and presents a standard deviation of 0.97 (Fig. 5). In contrast, if one attempts to estimate the probability distribution of $NQ_p$ conditioned to the occurrence of the 95[th] quantile of the normalized mean flow in June, no significant change is found in the estimate with respect to the unconditioned distribution. In fact, the resulting probability density function (pdf) for $NQ_p$ is centered around a mean value of 0.03 with a standard deviation of 0.998. The same behaviour is found in the probability distribution of the other dependent variable in its normalized form, $NQ_{mf}$, where the higher correlation coefficients (Table 3) determine even a greater displacement over the unconditioned distribution. In fact, the pdf of $NQ_{mf}$ conditioned to the occurrence of the 95[th] quantile value in the normalized mean flow in September is centered around a mean vaue of 0.64 and presents a standard deviation of 0.92 (Fig. 5).

In the upper Danube a similar scheme is found with the mean of the probability distribution of $NQ_p$ and $NQ_{mf}$ conditioned to the occurrence of the 95[th] quantile of the normalized mean flow in April, displaced





to 0.32 and 0.82 respectively (Fig. 6). The anomaly in the low correlation coefficient in March previously explained determines an insignificant change in the estimate with respect to the unconditioned distribution.

Figure 7 shows the comparison between the unconditioned flood frequency distribution and the updated

distributions in the untransformed domain when the flow in the previous month (September for the Po River, April for the Danube River) is higher than usual (70%, 80% and 95% quantile). For example, in the Po river, the unconditioned flood for a return period of 200 years, which results equal to 12507 $m^3 s^{-1}$, increases up to 13790 $m^3 s^{-1}$ (about 10% increase) when the mean flow in September corresponds to its 95% quantile. Similarly, in the upper Danube the unconditioned peak flow for a return period of

200 years, 10075 $m^3 s^{-1}$, increases up to 10861 $m^3 s^{-1}$ (about 8% increase) when the mean flow in April corresponds to its 95% quantile. The differences show that the average flow during the pre-flood seasons may indeed provide useful indications to update the flood frequency distribution.

## 5 Conclusions

The analysis of the observed mean daily flow values suggests the existence of LTP in both study sites

with H values above 0.71. Such persistence is exploited to improve streamflow forecasting in the flood season in terms of the mean monthly flow of the pre-flood seasons. To this end, we automatically detect the flood season through directional statistics and we fit a bivariate Gaussian distribution function to model the above dependence. A 10% and 8% increase in the 200-yr return period peak flows are found in the Po and Danube, respectively, when the average flows during the previous month corresponds to

its 95% quantile. The above results show that the Meta-Gaussian model applied to the streamflow records can be used for updating a season in advance the flood frequency distribution estimated for a given river, through a data assimilation approach by using the mean monthly flow of the pre-flood seasons.

The methodology herein proposed can be applied to any other study site once the parameters of the

Meta-Gaussian model confirm the presence of the above stochastic dependence. Also, other explanatory variables (e.g. rainfall, snowmelt, etc.) can be incorporated as long as there is stochastic dependence among the variables.





The findings presented in this paper highlight that river memory has an impact on flood formation and should then be properly considered for real time management of flood risk mitigation and resilience of societal settings to floods.

### Acknowledgements

The present work was (partially) developed within the framework of the Panta Rhei Research Initiative of the International Association of Hydrological Sciences (IAHS). Part of the results were elaborated in the Switch-On Virtual Water Science Laboratory, that was developed in the context of the project "SWITCH-ON" (Sharing Water-related Information to Tackle Changes in the Hydrosphere – for Operational Needs), funded by the European Union Seventh Framework Programme (FP7/2007-2013)

under grant agreement no. 603587. Cristina Aguilar acknowledges funding by both, the Juan de la Cierva Fellowship Programme of the Spanish Ministry of Economy and Competitiveness (JCI-2012-12802), and the José Castillejo Programme of the Spanish Ministry of Education, Culture and Sports (CAS14/00432).

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

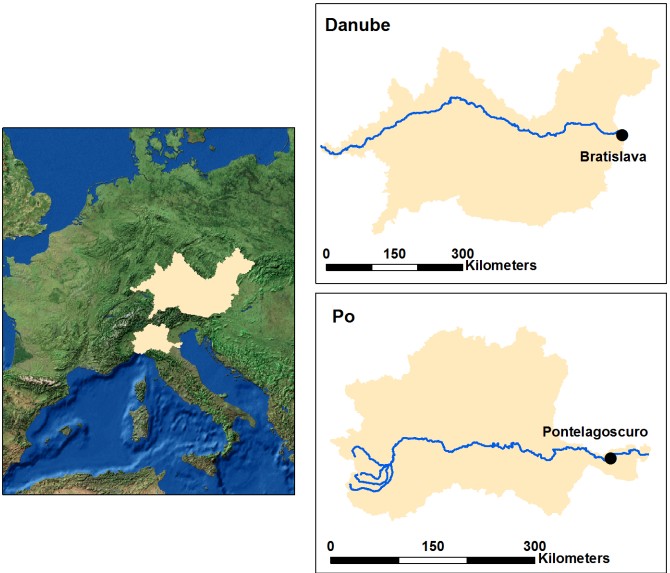

10   **Figure 1: Study sites. Danube river basin at Bratislava and Po river basin at Pontelagoscuro.**




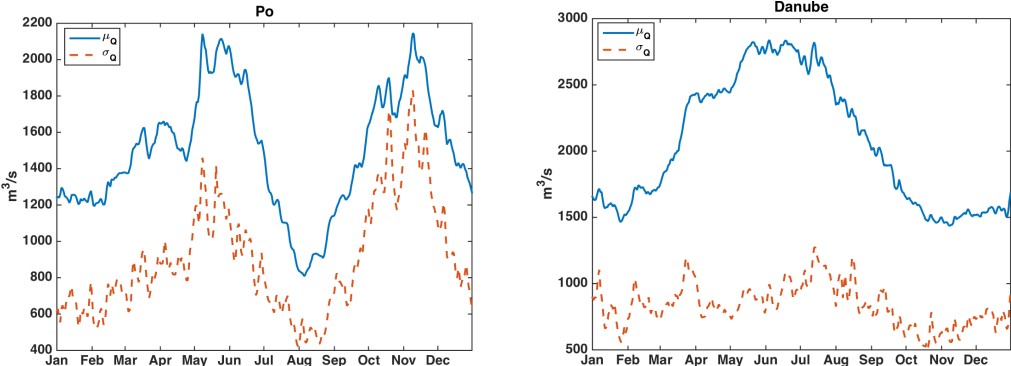

**Figure 2: Daily mean value $\mu_Q$ (m³s⁻¹) and daily standard deviation $\sigma_Q$ (m³s⁻¹) of the daily flows in the observation periods: 1920-2009 in the Po at Pontelagoscuro, 1901-2007 in the Danube at Bratislava**

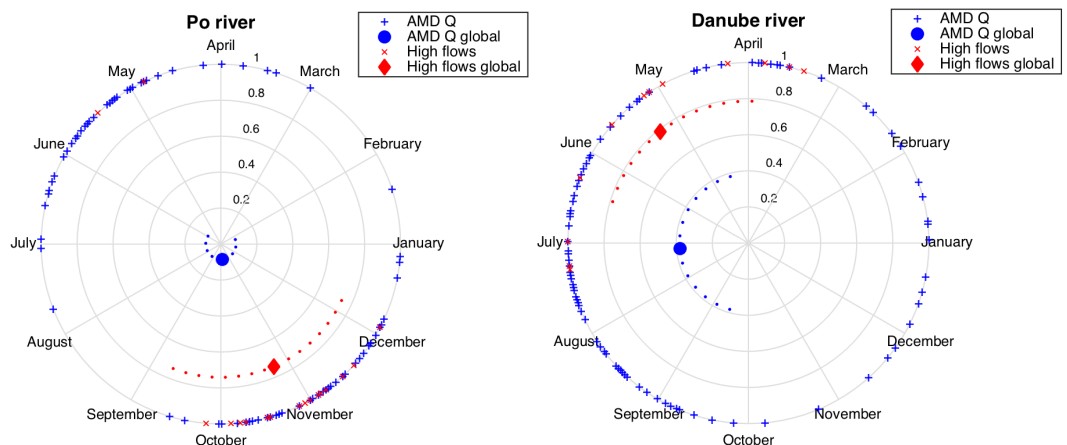

**Figure 3. Seasonality space representation of the annual maximum daily flows (AMD) and high flow events. Dots around the global value indicate the dispersion.**




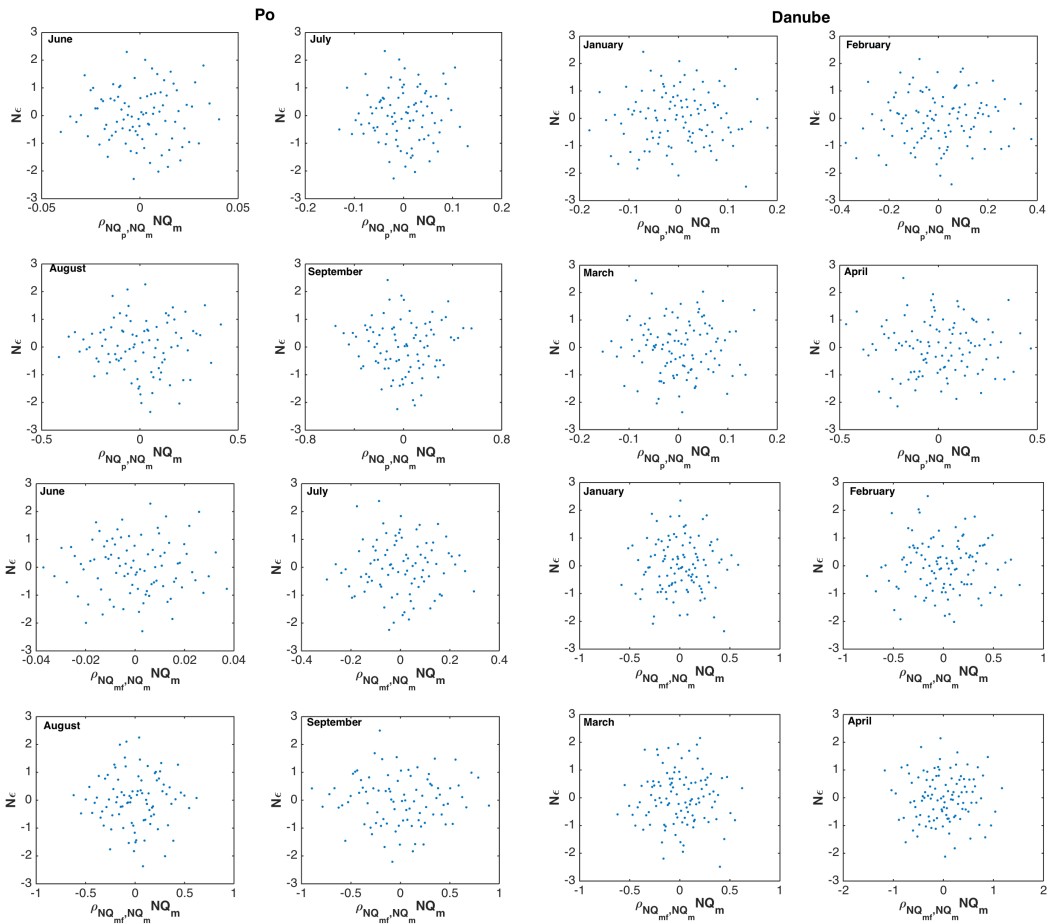

**Figure 4. Residual plot of the linear regression of NQ$_m$ on NQ$_p$ and NQ$_{mf}$ in the Po river (left) and upper Danube (right)**





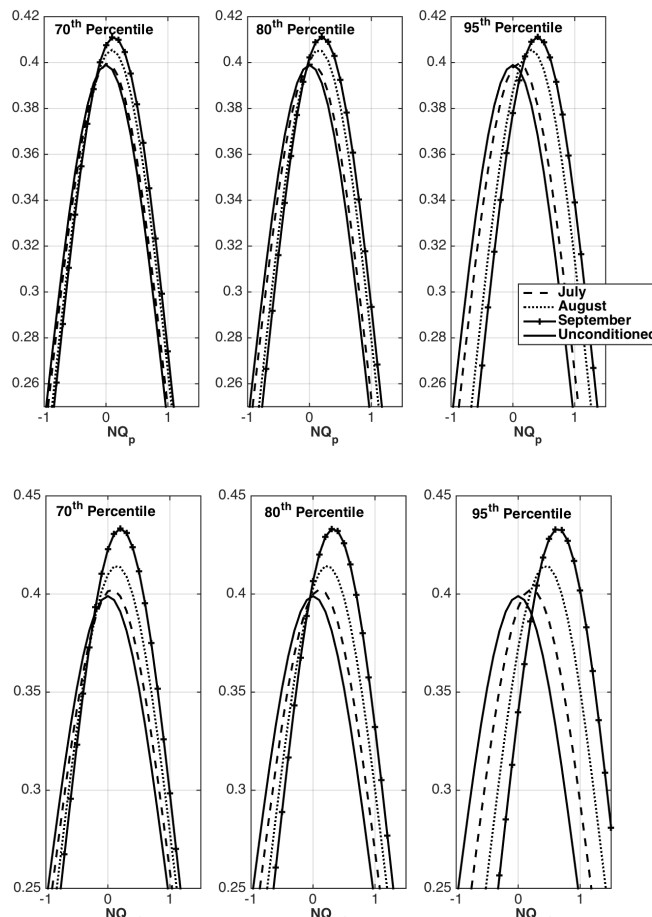

**Figure 5. Probability distribution functions of the normalized dependent variables (NQ$_p$ and NQ$_{mf}$) conditioned to the occurrence of the 70$^{th}$, 80$^{th}$ and 95$^{th}$ percentiles of the normalized variables in the pre-flood season in the Po river.**





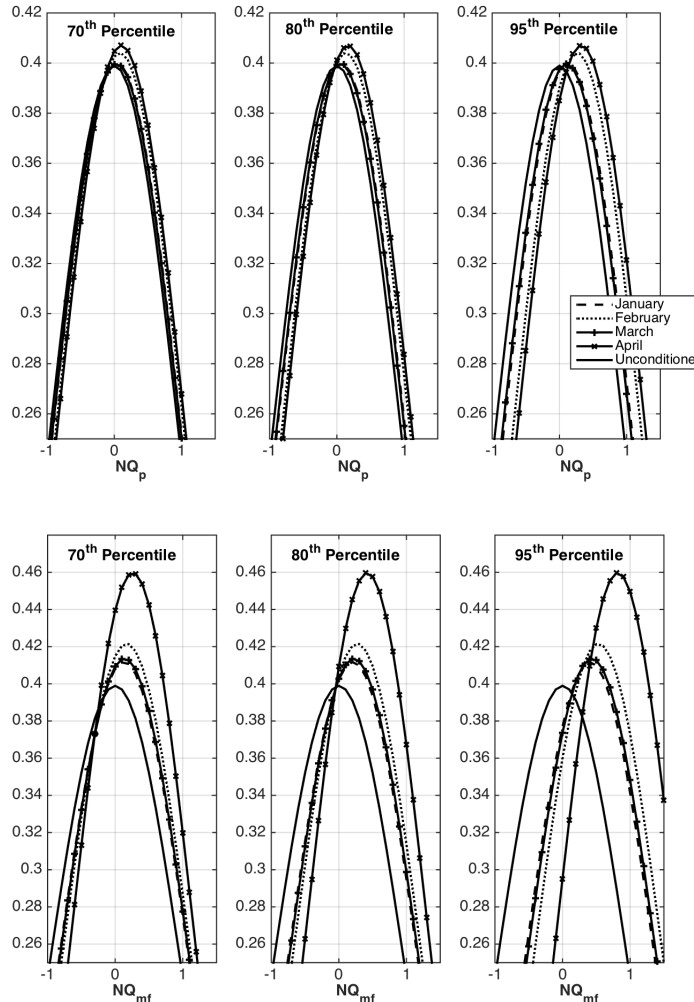

**Figure 6. Probability distribution functions of the normalized dependent variables (NQ$_p$ and NQ$_{mf}$) conditioned to the occurrence of the 70$^{th}$, 80$^{th}$ and 95$^{th}$ percentiles of the normalized variables in the pre-flood season in the upper Danube.**





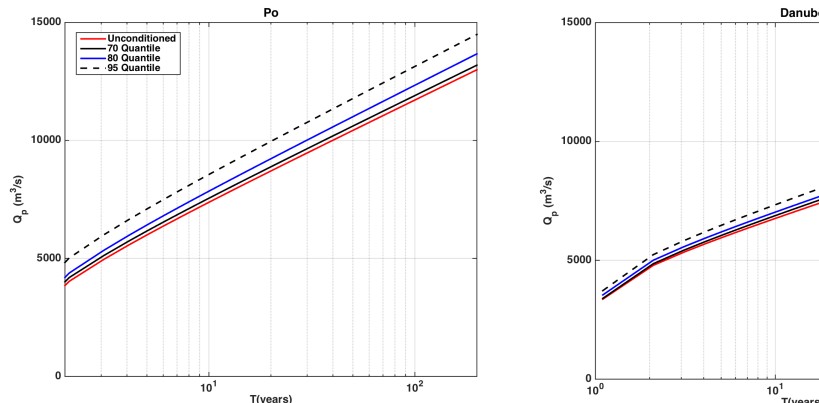

**Figure 7. Peak flows in the flood season (Oct-Nov in the Po, May-July in the upper Danube) *vs* return period modeled through the EV1 distribution function.**





**Table 1. Data description of observed time series. Descriptive statistics are given for non-deseasonalized data.**

|  | Po | Danube |
|---|---|---|
| *Observation period* | 1920-2009 | 1901-2007 |
| *Daily discharge* | | |
| Gauge location | Pontelagoscuro | Bratislava |
| Catchment area ($km^2$) | 71 000 | 131 331 |
| Mean ($m^3\ s^{-1}$) | 1470 | 2053 |
| Standard deviation ($m^3\ s^{-1}$) | 1007 | 973 |
| Fluvial regime | Pluvial regime. Two peak periods | Alpine regime. One peak in the summer |
| *Monthly precipitation* | | |
| Number of weather stations | 18 | 16 |
| Mean (mm/month) | 72 | 73 |
| Standard deviation (mm/month) | 17 | 37 |
| *Monthly temperature* | | |
| Number of weather stations | 12 | 11 |
| Mean (°C) | 12.9 | 7.9 |
| Standard deviation (°C) | 7.5 | 7.2 |





**Table 2. Estimated H values on deseasonalized data series applying R/S statistic (R/S), Aggregated Variance Method (AV), and Differenced Variance Method (DV)**

|  | R/S | AV | DV |
|---|---|---|---|
| *Po (Pontelagoscuro)* | | | |
| Daily Q | 0.81 | 0.74 | 0.94 |
| Monthly Q | 0.76 | 0.62 | 0.80 |
| Monthly P | 0.61 | 0.59 | 0.60 |
| Monthly T | 0.64 | 0.80 | 0.90 |
| | | | |
| *Danube (Bratislava)* | | | |
| Daily Q | 0.80 | 0.71 | 0.86 |
| Monthly Q | 0.75 | 0.54 | 0.79 |
| Monthly P | 0.56 | 0.36 | 0.56 |
| Monthly T | 0.61 | 0.76 | 0.70 |

**Table 3. Pearson's cross correlation coefficient between both, $NQ_p$ and $NQ_{mf}$, and $NQ_m$ for varying antecedent monthly flow. Flood season in Po: October-November. Flood season in Danube: May-July**

| Po | | | Danube | | |
|---|---|---|---|---|---|
| Month | $\rho(NQ_m, NQ_p)$ | $\rho(NQ_m, NQ_{mf})$ | Month | $\rho(NQ_m, NQ_p)$ | $\rho(NQ_m, NQ_{mf})$ |
| September | 0.24 | 0.39 | April | 0.20 | 0.50 |
| August | 0.18 | 0.27 | March | 0.06 | 0.26 |
| July | 0.06 | 0.13 | February | 0.16 | 0.32 |
| June | 0.02 | -0.02 | January | 0.07 | 0.25 |
| May | -0.06 | -0.05 | December | -0.002 | 0.17 |
| April | -0.13 | -0.07 | November | 0.05 | 0.09 |
| March | -0.18 | -0.12 | October | 0.13 | 0.10 |
| February | -0.04 | -0.05 | September | -0.07 | -0.08 |
| January | -0.07 | -0.07 | August | -0.21 | 0.09 |