# Peer review of "Real time updating of the flood frequency distribution through data assimilation"

_Hydrology and Earth System Sciences, 2016_

## Referee Comment (RC1) · Anonymous Referee #1 · 16 Jan 2017

The manuscript describes an analysis to update the flood frequency distribution including some proxies of the catchment memory.

The topic is interesting and the manuscript is well written and pleasant to read.

I have just few comments and suggestions to share with the authors.

Authors adopted the Normal Quantile Transformation in order to build and manage a bivariate Gaussian distribution, however they did not justify this choice. My feeling is that a bivariate distribution directly inferenced on the data could provide more accurate results.

It would be interesting to quantify the uncertainty of the proposed conditioned distributions.

Minor point: page 11 line 14. It should be Table 3.

---

## Referee Comment (RC2) · Anonymous Referee #2 · 26 Jan 2017

Main points

This paper aims to assess whether the use of pre-flood season streamflow data can help forecast the probability of high floods. The paper is well-written, gives a nice review of previous literature, and presents a clear goal. The topic has important scientific contributions and societal implications (ability to improve flood forecasting).

My main comment is that the study could benefit from adding a cross-validation of the method. The current results seem to largely be a description of the pattern found for these two rivers. Adding cross validation could illustrate how identifying this relationship can help inform flood forecasting, as well as demonstrate the utility of this method. For example, what if you select years with anomalously high flows, omit them from the fitting procedure, and then assess how much this method improves prediction of floods

in these years?

Also, it would be good to provide more rationale for Normal Quantile Transform as this choice likely has a big impact on results since floods generally aren't normally distributed. In addition, since meta-Gaussian models aren't commonly used it hydrology it would be helpful to add some more background on them. For example, why don't these models have fitted slope coefficients? Are these models considered linear models? Are the cross-correlation coefficients modified to ensure residuals have a mean of zero? Readers with a more traditional statistics background will likely be looking for these components of the models.

You present a review of LTP in the introduction and the abstract includes that the approach assumes flood formation is driven in part by "long term perturbations". Usually I think of long term as referring to longer than a year, but later you define "long term stress, like higher than usual rainfall lasting for several months". Can you explain a bit more about how you are defining "long term" and the link with LTP given that 9 months before flood season is the farthest back you look at correlation? And that flows before flood season only have positive correlation with during flood season for the Po river for preceding 3-4 months.

I am also curious how looking at shorter record length impacts the correlation between these variables? The data was de-trended and de-seasonalized - does that mean that the correlations shown in Table 3 are stable even for subsets of the whole record you have for these rivers?

More explanation of how to interpret figure 3 would be good for those of us not familiar with these types of figures. Do the regularity values come from which concentric circle the points fall on?

Model residuals appear homoscedastic but what about normality? Perhaps you can mention that meta-Gaussian models don't require the usual assumption about lack of correlation in residuals, right?

It was unclear to me why was temperature was included in the study. Temperature patterns are discussed in the section on long term persistence, but there isn't an explanation of how temperature relates to the model or goals of the study.

The conclusion section is quite short and could benefit from some additional discussion, perhaps something about the utility of the method for other locations and differing catchment sizes. The abstract notes "The proposed technique may allow one to reduce the uncertainty associated to the estimation of flood frequency" – could you could elaborate on this in the conclusion?

Minor comments

The use of the word "significant" should be clarified (as in the abstract and p 12, line 19). Do you mean statistical significance? At what level?

Abstract Lines 15-17: I think this would be clearer if re-organized, perhaps: "To exploit the above sensitivity to long term perturbations, a Meta-Gaussian model and a data assimilation approach is implemented for updating the flood frequency distribution a season in advance."

Abstract Line 20: A word is missing: I would suggest adding "which" before "occurred (ie, "which occurred" or even "occurring" rather than just "occurred")

P 2 Line 11: an "a" is missing before "long time"

P 2 Line 15: "associated with" more commonly used than ""associated to"

P 4 Line 20 Extra word "on" before "the trend"

P 4 Line 24 I suggest adding a comma after "As stated in the Danube River Basin Management Plan" (since it is a dependent clause)

P 4 Line 25 significantly rather than significant?

Page 9 line 18 "was" after a plural sounds strange – perhaps "We applied directional

statistics..."

P 10, lines 4 This makes it sound like 0.71 is a sort of cut off. Perhaps "H values above 0.5" would be better here, reminding us that that is the cut-off of interest. Or you could say "H values of 0.71 or higher".

P 11 line 14 You refer to table 2 but I believe you mean table 3

P 11 line 15 I'm not sure what you mean by "we appreciate" here

P 12 line 1 Is it really "A goodness-of-fit test" or more an evaluation that model assumptions about the residuals are met?

P 12 line 13 – again, perhaps "we find" rather than "we appreciate"

P 12 line 19 June is mentioned but not shown on the plots? Was that intentional?

P 13 lines 1-3 The text has: "The anomaly in the low correlation coefficient in March previously explained determines an insignificant change in the estimate with respect to the unconditioned distribution." But it appears that the line corresponding to march coincides with the line corresponding to January, not to the unconditioned distribution.

Figure 4 – I find it helpful to add a horizontal line at 0 when assessing homoscedasticity of residuals. Though I'm not sure we need to see these plots. Just describing them in the text is probably sufficient.

Figures 5-6 Adding a-f markers to each subplot would help with finding the plot being discussed in the text; add labels to y-axes

Figure 7 add to caption that the quantiles refer to flows higher than usual in the previous month.

[Figure]

---

## Author Comment (AC1) · 9 Mar 2017

**Real time updating of the flood frequency distribution through data assimilation**

Reply to the reviewers' comments

We are extremely grateful to the reviewers for their valuable comments and we would also like to thank the Editor and the Editorial Office for the assistance provided during the review process.

**Reply to reviewer #1**
Reviewer #1 is satisfied by the paper in the present form and provides a minor comment. He/she suggested justifying the choice of the NQT over a bivariate distribution directly inferenced on the data as he/she feels that a bivariate distribution directly inferenced from the data could provide more accurate results. Also he/she suggests quantifying the uncertainty of the proposed conditioned distributions.

Accordingly, we added the following statement to justify the choice of the method in section 3.2 in the revised version: "Among the advantages of the NQT, reviewed by Maranzano and Krzysztofowicz (2004), emerges the fact that it is free of any distributional assumption. Thus, the NQT allows one to avoid the selection of a suitable parametric model for the distribution of the considered hydrological variable."

As for quantification of uncertainty, we agree that this is an important issue that deserves to be discussed in the revised paper and therefore we will include an extensive discussion, which we anticipate here below.

Uncertainty in the conditioned distributions is mainly given by two sources: the first is uncertainty in the NQT, namely, uncertainty in the estimation of the marginal probability distribution of independent and dependent variables in the regression. The NQT is a non-parametric transformation and therefore its uncertainty cannot be determined quantitatively (see Maranzano and Krzytofowicz, 2004) and Montanari and Brath, 2004). We will emphasize that it is advisable that NQT is estimated by using long records encompassing a wide range of possible meteorological and hydrological conditions.

The second source of uncertainty is related to the estimation of the cross correlation coefficient between dependent and independent variables in the Gaussian domain. This uncertainty can be quantified for a given confidence level and again depends on the length of the records. We will determine such uncertainty quantitatively in the revised version of the paper.

**Reply to reviewer #2**
Reviewer #2 is also satisfied by the paper but raises some issues and provides several comments in order to strengthen the value of the study. We offer a detailed point-by-point discussion here below where the original comments by the reviewer are copied in italics.

*My main comment is that the study could benefit from adding a cross-validation of the method. The current results seem to largely be a description of the pattern found for these two rivers. Adding cross validation could illustrate how identifying this relationship can help inform flood forecasting, as well as demonstrate the utility of this method. For example, what if you select years with anomalously high flows, omit them from the fitting procedure, and then assess how much this method improves prediction of floods in these years?*

We agree that a cross-validation in both rivers would definitely demonstrate the utility of the method regardless of the study site. This analysis will be included in the revised version. Additional paragraphs will be included in both the methodology (section 3.2) and the results section (section 4.2.2). We provide here an anticipation of the results.

We carried out a cross-validation of the method by removing the year with the wettest pre-flood season (1977 in the Po and 1944 in the Danube). We noticed a very little change in the cumulative distribution functions, which is a first demonstration of the robustness of the method (Fig. 1). Afterwards, we simulated a real time prediction of the probability distribution of the flood flows in the next flood season (Fig. 2). Results will be presented in the revised version of the manuscript.

[Figure]

Figure 1. Cumulative distribution functions of the predicted $Q_p$ in terms of the $Q_m$ value in 1977 with the NQT considering both the whole data set (calibration set); and removing the data in 1977 (in the Po) and 1944 (in the Danube) from the analysis (validation set). The observed value is highlighted in blue.

[Figure]

Figure 2. Probability distribution of the predicted $Q_p$ in terms of the $Q_m$ value in 1977 with the NQT considering both the whole data set (calibration set); and removing the data in 1977 (in the Po) and 1944 (in the Danube) from the analysis (validation set). The observed value is highlighted in blue.

*Also, it would be good to provide more rationale for Normal Quantile Transform as this choice likely has a big impact on results since floods generally aren't normally distributed. In addition, since meta-Gaussian models aren't commonly used it hydrology it would be helpful to add some more background on them. For example, why don't these models have fitted slope coefficients? Are these models considered linear models? Are the cross-correlation coefficients modified to ensure residuals have a*

*mean of zero? Readers with a more traditional statistics background will likely be looking for these components of the models.*

Following the reviewer's suggestion, we will make reference to the literature in which a full description of the NQT can be found (Krzysztofowicz, 1997; Kelly and Krzysztofowicz, 1997). Besides, we will add some more references to previous applications of the NQT in hydrological studies: Krzysztofowicz and Kelly (2000); Krzysztofowicz and Herr (2001); Krzysztofowicz and Maranzano (2004a, b); Maranzano and Krzysztofowicz (2004). Also, we will add the following paragraph in order to justify further the choice of the method: "Among the advantages of the NQT, reviewed by Maranzano and Krzysztofowicz (2004), emerges the fact that it is free of any distributional assumption. Thus, the NQT allows one to avoid the selection of a suitable parametric model for the distribution of the considered hydrological variable". These modifications will be incorporated in section 3.2 in the revised version.

*+why don't these models have fitted slope coefficients?*

Because they just need the standard normal distribution and the marginal probability distribution to define the NQT of the original variates.

*+Are these models considered linear models?*

Yes they are, as already stated in the paragraph previous to equation 7:

$$NQ_p(t) = \rho(\mathbf{NQ_m}, \mathbf{NQ_p})NQ_m(t) + N\varepsilon(t) \tag{7}$$

*+Are the cross-correlation coefficients modified to ensure residuals have a mean of zero?*

Cross-correlation coefficients are not modified, but the linear model assumes that residuals ($\varepsilon$) are considered to be an outcome of an stochastic process, which is independent, homoscedastic, stochastically independent of $\mathbf{NQ_m}$, and normally distributed with zero mean and variance given by $1-\rho^2(\mathbf{NQ_m}, \mathbf{NQ_p})$. The goodness of fit based on the behavior of the residuals verifies that these conditions are met and therefore ensures that residuals have a mean of zero.

*You present a review of LTP in the introduction and the abstract includes that the approach assumes flood formation is driven in part by "long term perturbations". Usually I think of long term as referring to longer than a year, but later you define "long term stress, like higher than usual rainfall lasting for several months". Can you explain a bit more about how you are defining "long term" and the link with LTP given that 9 months before flood season is the farthest back you look at correlation? And that flows before flood season only have positive correlation with during flood season for the Po river for preceding 3-4 months.*

Unlike short term perturbations, that we consider to be driven by short term meteorological forcings leading to infiltration and/or saturation excess, long term perturbations may be due to higher-than-usual storage in the catchment, which may cause the presence of seasonal to interannual correlation.

We agree with the reviewer that long term persistence, which we mathematically define in the manuscript, is in principle extended to the whole past history of the process and therefore not only to the few past seasons. We are also aware that seasonal correlation is not necessarily caused by LTP, as it may also be originated by short term correlation. However, the presence of LTP makes seasonal correlation more likely and therefore we believe that a study focusing on seasonal correlation should be supported by an estimation of the possible presence of LTP. This is the reason why we believe that LTP estimation fits in our paper. We will expand this discussion in the revised version of the manuscript.

*I am also curious how looking at shorter record length impacts the correlation between these variables? The data was de-trended and deseasonalized - does that mean that the correlations shown in Table 3 are stable even for subsets of the whole record you have for these rivers?*

It is well known that correlations computed on short records may become unreliable. We agree that this is an important issue and therefore we will include in the revised version of the paper a quantitative estimation of uncertainty for the cross correlation coefficient.

For our specific case, the data sets considered in our study are very long and therefore we are convinced that uncertainty in the estimation of the cross-correlation coefficient is negligible. The minimum data length to ensure meaningful conclusions depends on the statistical behaviors of the time series and, in particular, variability, as we will discuss in the revised version of the paper. From a hydrological point of view, it is important that records are long and encompass a wide range of possible meteorological and hydrological conditions.

To make an experiment, we took 100 random subsets of 70 years in Po and 85 years in the Danube from the original datasets and we computed the correlation coefficients for each subset. As an example, Figure 2 represents the results for the peak flows in both rivers whose basic statistics are shown in Table 1. As we can see, we obtained a mean value similar to those obtained with the whole data sets (Table 3 in the manuscript), and the standard deviation just changes the second decimal digit.

[Figure]

Figure 2. Pearson's cross correlation coefficient between $NQ_p$ and $NQ_{mf}$ for 100 random trials of subsets in the original datasets. Flood season in Po: October-November. Flood season in Danube: May-July

Table 1. Basic statistics of the Pearson's cross correlation coefficient between $NQ_p$ and $NQ_{mf}$ for 100 random trials of subsets in the original datasets.

| | $\rho(NQ_m, NQ_p)$ | |
| --- | --- | --- |
| | **Po** | **Danube** |
| Mean | 0.25 | 0.20 |
| Maximum | 0.40 | 0.32 |
| Minimum | 0.16 | 0.08 |
| Standard deviation | 0.05 | 0.05 |

Finally, we would like to emphasize that data were detrended and deseasonalized for long-term memory assessments but the analysis of the statistical dependence between the peak flow in the flood season and the average flow in the pre-flood season was carried out with the original data set. Nevertheless, we also computed the correlations with the detrended and deseasonalized series and they do not change much (just the third decimal digit). We found a higher change in the correlation values (but still not a great change, just variations in the second decimal digit) when we applied the cross-correlation and removed the wettest year in the pre-flood season.

*More explanation of how to interpret figure 3 would be good for those of us not familiar with these types of figures. Do the regularity values come from which concentric circle the points fall on?*

In order to better interpret figure 3, the following paragraph will be added at the end of section 3: "Results are shown in a circle plot where each date of occurrence of the variables analyzed in the data set is visible along the perimeter. The timing of the global occurrence of each of the variables analyzed can be easily identified in terms of the slice of month where it falls. Also, the proximity to the center of the circle indicates the regularity of the phenomenon, with the highest regularity found in the perimeter of the circle."

*Model residuals appear homoscedastic but what about normality? Perhaps you can mention that meta-Gaussian models don't require the usual assumption about lack of correlation in residuals, right?*

Residuals were also checked for normality as this is an underlying assumption. The manuscript will be revised to make this requirement clear.

Serial correlation is assumed to be negligible, as we are regressing the annual peak flow against the average flow in the low flow season. Therefore, each year we have a pair of data and thus the above assumption is fully justified and in our opinion does not need to be checked (see also Montanari and Brath, 2004; Maranzano and Krzystofowicz, 2004).

*It was unclear to me why was temperature was included in the study. Temperature patterns are discussed in the section on long term persistence, but there isn't an explanation of how temperature relates to the model or goals of the study.*

Temperature as well as rainfall were included to verify whether the hypothesis of the presence of LTP is supported by data evidence in both rivers. We consider both variables to be the major drivers of river flows in both rivers, where both water mass and energy balances determine the response of the catchment. Once we demonstrated that the presence of LTP in the river flow data may be a reasonable working assumption, we use the river flow in the pre-flood season as explanatory variable. We provided in the introduction of the paper a justification for the use of river flow data as a proxy for mean areal rainfall over the catchment.

*The conclusion section is quite short and could benefit from some additional discussion, perhaps something about the utility of the method for other locations and differing catchment sizes. The abstract notes "The proposed technique may allow one to reduce the uncertainty associated to the estimation of flood frequency" – could you could elaborate on this in the conclusion?*

We agree with the reviewer and will extend the concluding remarks by discussing the applicability of the method to other cases.

*Minor comments*

*The use of the word "significant" should be clarified (as in the abstract and p 12, line 19). Do you mean statistical significance? At what level?*

We used the term "significant" to mean "noticeable". We will change the terminology to avoid confusion with terms used in statistics.

*Abstract Lines 15-17: I think this would be clearer if re-organized, perhaps: "To exploit the above sensitivity to long term perturbations, a Meta-Gaussian model and a data assimilation approach is implemented for updating the flood frequency distribution a season in advance."* OK

*Abstract Line 20: A word is missing: I would suggest adding "which" before "occurred (ie, "which occurred" or even "occurring" rather than just "occurred")* OK

*P 2 Line 11: an "a" is missing before "long time"* OK

*P 2 Line 15: "associated with" more commonly used than ""associated to"* OK

*P 4 Line 20 Extra word "on" before "the trend"* OK

*P 4 Line 24 I suggest adding a comma after "As stated in the Danube River Basin Management Plan" (since it is a dependent clause)* OK

*P 4 Line 25 significantly rather than significant?* OK

*Page 9 line 18 "was" after a plural sounds strange – perhaps "We applied directional statistics. . ."* OK

*P 10, lines 4 This makes it sound like 0.71 is a sort of cut off. Perhaps "H values above 0.5" would be better here, reminding us that that is the cut-off of interest. Or you could say "H values of 0.71 or higher".* OK

*P 11 line 14 You refer to table 2 but I believe you mean table 3*

Yes, we apologize for the error. It will be corrected in the revised version.

*P 11 line 15 I'm not sure what you mean by "we appreciate" here.*

We changed it to "always found"

*P 12 line 1 Is it really "A goodness-of-fit test" or more an evaluation that model assumptions about the residuals are met?*

We agree that it is an evaluation that the model assumptions about the residuals are met. We will revise the text accordingly.

*P 12 line 13 – again, perhaps "we find" rather than "we appreciate"* OK

*P 12 line 19 June is mentioned but not shown on the plots? Was that intentional?*

We regret that there was an error and we meant July. It has been corrected in the revised version.

*P 13 lines 1-3 The text has: "The anomaly in the low correlation coefficient in March previously explained determines an insignificant change in the estimate with respect to the unconditioned distribution." But it appears that the line corresponding to march coincides with the line corresponding to January, not to the unconditioned distribution.*

Yes, we apologize for the error. It has been corrected in the revised version.

*Figure 4 – I find it helpful to add a horizontal line at 0 when assessing homoscedasticity of residuals. Though I'm not sure we need to see these plots. Just describing them in the text is probably sufficient.*

We regret to disagree here with the reviewer as these plots show the evaluation of the behavior of the residuals and so, including them allows us to assume that the linear model proposed can be applied to our data.

*Figures 5-6 Adding a-f markers to each subplot would help with finding the plot being discussed in the text; add labels to y-axes* OK

*Figure 7 add to caption that the quantiles refer to flows higher than usual in the previous month.* OK

---

## Author Response (AR1)

**Real time updating of the flood frequency distribution through data assimilation**

Reply to the reviewers' comments

We are extremely grateful to the reviewers for their valuable comments and we would also like to thank the Editor and the Editorial Office for the assistance provided during the review process.

**Reply to reviewer #1**

Reviewer #1 is satisfied by the paper in the present form and provides a minor comment. He/she suggested justifying the choice of the NQT over a bivariate distribution directly inferenced on the data as he/she feels that a bivariate distribution directly inferenced from the data could provide more accurate results. Also he/she suggests quantifying the uncertainty of the proposed conditioned distributions.

Accordingly, we added the following statement to justify the choice of the method in section 3.2 in the revised version (page 7): "Among the advantages of the NQT, reviewed by Maranzano and Krzysztofowicz (2004), emerges the fact that it is free of any distributional assumption. Thus, the NQT allows one to avoid the selection of a suitable parametric model for the distribution of the considered hydrological variable."

As for quantification of uncertainty, we agree that this is an important issue that deserved to be discussed in the revised paper and therefore we included an extensive discussion in section 3.2 (pages 8 and 9) which is summarized here below.

Uncertainty in the conditioned distributions is mainly given by two sources: the first is uncertainty in the NQT, namely, uncertainty in the estimation of the marginal probability distribution of independent and dependent variables in the regression. The NQT is a non-parametric transformation and therefore its uncertainty cannot be determined quantitatively (see Maranzano and Krzytofowicz, 2004) and Montanari and Brath, 2004). We emphasize that it is advisable that NQT is estimated by using long records encompassing a wide range of possible meteorological and hydrological conditions.

The second source of uncertainty is related to the estimation of the cross-correlation coefficient between dependent and independent variables in the Gaussian domain. This uncertainty can be quantified for a given confidence level and again depends on the length of the records. We propose a procedure to estimate such uncertainty quantitatively and present an application in the revised version of the paper (Section 4.2.3, page 14 and Figure 8).

**Reply to reviewer #2**

Reviewer #2 is also satisfied by the paper but raises some issues and provides several comments in order to strengthen the value of the study. We offer a detailed point-by-point discussion here below where the original comments by the reviewer are copied in italics.

*My main comment is that the study could benefit from adding a cross-validation of the method. The current results seem to largely be a description of the pattern found for these two rivers. Adding cross validation could illustrate how identifying this relationship can help inform flood forecasting, as well as demonstrate the utility of this method. For example, what if you select years with*

*anomalously high flows, omit them from the fitting procedure, and then assess how much this method improves prediction of floods in these years?*

We agree that a validation (in the revised paper we use the term leave-one-out validation instead of cross validation) in both rivers would definitely demonstrate the utility of the method regardless of the study site. This analysis has been included in the revised version. Additional paragraphs have been included in both the methodology (section 3.2, pages 8-9) and the results section (section 4.2.3, page 14). We provide here an anticipation of the results.

We removed the year with the wettest pre-flood season (1977 in the Po and 1944 in the Danube) and computed the correlation coefficients as well as their 95% confidence bands as explained in section 3.2. Then, we estimated the probability distribution for the peak flow in the flood season for that year. Figure 1 (Figure 8 in the revised manuscript) shows the unconditioned flood frequency distribution along with the updated one as well as the 95% confidence bands for the latter.

[Figure]

**Figure 1. Leave-one-out cross validation. Unconditioned EV1 probability distribution of peak flows for the year with the wettest pre-flood season (1977 in the Po, 1944 in the upper Danube) along with conditioned distributions with related 95% confidence bands.**

*Also, it would be good to provide more rationale for Normal Quantile Transform as this choice likely has a big impact on results since floods generally aren't normally distributed. In addition, since meta-Gaussian models aren't commonly used it hydrology it would be helpful to add some more background on them. For example, why don't these models have fitted slope coefficients? Are these models considered linear models? Are the cross-correlation coefficients modified to ensure residuals have a mean of zero? Readers with a more traditional statistics background will likely be looking for these components of the models.*

Following the reviewer's suggestion, we made reference to the literature in which a full description of the NQT can be found (Krzysztofowicz, 1997; Kelly and Krzysztofowicz, 1997). Besides, we added more references to previous applications of the NQT in hydrological studies: Krzysztofowicz and Kelly (2000); Krzysztofowicz and Herr (2001); Krzysztofowicz and Maranzano (2004a, b); Maranzano and Krzysztofowicz (2004). Also, we added the following paragraph in order to justify further the choice of the method: "Among the advantages of the NQT, reviewed by Maranzano and Krzysztofowicz (2004), emerges the fact that it is free of any distributional assumption. Thus, the NQT allows one to avoid the selection of a suitable parametric model for the distribution of the considered hydrological variable". These modifications are incorporated in section 3.2 in the revised version (pages 6-7).

*+why don't these models have fitted slope coefficients?*

Because they just need the standard normal distribution and the marginal probability distribution to define the NQT of the original variates.

+*Are these models considered linear models?*

Yes they are, as already stated in the paragraph previous to equation 2:

$$NQ_p(t) = \rho(\mathbf{NQ_m}, \mathbf{NQ_p})NQ_m(t) + N\varepsilon(t) \tag{2}$$

+*Are the cross-correlation coefficients modified to ensure residuals have a mean of zero?*

Cross-correlation coefficients are not modified, but the linear model assumes that residuals ($\varepsilon$) are considered to be an outcome of an stochastic process, which is independent, homoscedastic, stochastically independent of $\mathbf{NQ_m}$, and normally distributed with zero mean and variance given by $1-\rho^2(\mathbf{NQ_m}, \mathbf{NQ_p})$. The evaluation based on the behavior of the residuals verifies that these conditions are met and therefore ensures that residuals have a mean of zero.

*You present a review of LTP in the introduction and the abstract includes that the approach assumes flood formation is driven in part by "long term perturbations". Usually I think of long term as referring to longer than a year, but later you define "long term stress, like higher than usual rainfall lasting for several months". Can you explain a bit more about how you are defining "long term" and the link with LTP given that 9 months before flood season is the farthest back you look at correlation? And that flows before flood season only have positive correlation with during flood season for the Po river for preceding 3-4 months.*

Unlike short term perturbations, that we consider to be driven by short term meteorological forcings leading to infiltration and/or saturation excess, long term perturbations may be due to higher-than-usual storage in the catchment, which may cause the presence of seasonal to interannual correlation.

We agree with the reviewer that long term persistence, which we mathematically define in the manuscript, is in principle extended to the whole past history of the process and therefore not only to the few past seasons. We are also aware that seasonal correlation is not necessarily caused by LTP, as it may also be originated by short term correlation. However, the presence of LTP makes seasonal correlation more likely and therefore we believe that a study focusing on seasonal correlation should be supported by an estimation of the possible presence of LTP. This is the reason why we believe that LTP estimation fits in our paper. We expanded this discussion in the revised version of the manuscript.

*I am also curious how looking at shorter record length impacts the correlation between these variables? The data was de-trended and deseasonalized - does that mean that the correlations shown in Table 3 are stable even for subsets of the whole record you have for these rivers?*

It is well known that correlations computed on short records may become unreliable. We agree that this is an important issue and therefore we included in the revised version of the paper a quantitative estimation of uncertainty for the cross-correlation coefficient (Table 3).

For our specific case, the data sets considered in our study are very long and therefore we are convinced that uncertainty in the estimation of the cross-correlation coefficient is negligible. The minimum data length to ensure meaningful conclusions depends on the statistical behaviors of the time series and, in particular, variability, as we discussed in the revised version of the paper. From a hydrological point of view, it is important that records are long and encompass a wide range of possible meteorological and hydrological conditions.

To make an experiment, we took 100 random subsets of 70 years in Po and 85 years in the Danube from the original datasets and we computed the correlation coefficients for each subset. As an example, Figure 2 represents the results for the peak flows in both rivers whose basic statistics are shown in Table 1. As we can see, we obtained a mean value similar to those obtained with the whole data sets (Table 3 in the manuscript), and the standard deviation just changes the second decimal digit.

[Figure]

Figure 2. Pearson's cross-correlation coefficient between $NQ_p$ and $NQ_{mf}$ for 100 random trials of subsets in the original datasets. Flood season in Po: October-November. Flood season in Danube: May-July

Table 1. Basic statistics of the Pearson's cross-correlation coefficient between $NQ_p$ and $NQ_{mf}$ for 100 random trials of subsets in the original datasets.

| $\rho(NQ_m, NQ_p)$ | | |
|---|---|---|
| | **Po** | **Danube** |
| Mean | 0.25 | 0.20 |
| Maximum | 0.40 | 0.32 |
| Minimum | 0.16 | 0.08 |
| Standard deviation | 0.05 | 0.05 |

Finally, we would like to emphasize that data were detrended and deseasonalized for long-term memory assessments but the analysis of the statistical dependence between the peak flow in the flood season and the average flow in the pre-flood season was carried out with the original data set. Nevertheless, we also computed the correlations with the detrended and deseasonalized series and they do not change much (just the third decimal digit). We found a higher change in the correlation values (but still not a great change, just variations in the second decimal digit) when we applied the leave-one-out correlation and removed the wettest year in the pre-flood season.

*More explanation of how to interpret figure 3 would be good for those of us not familiar with these types of figures. Do the regularity values come from which concentric circle the points fall on?*

In order to better interpret figure 3, the following paragraph has been added at the end of section 3: "Results are shown in a circle plot where each date of occurrence of the variables analyzed in the data set are visible along the perimeter. The month of occurrence of each of the variables can be easily identified. Also, the proximity to the center of the circle of the global value indicates the regularity of the phenomenon with the highest regularity found in the perimeter of the circle."

*Model residuals appear homoscedastic but what about normality? Perhaps you can mention that meta-Gaussian models don't require the usual assumption about lack of correlation in residuals, right?*

Residuals were also checked for normality as this is an underlying assumption. The manuscript has been revised to make this requirement clear.

Serial correlation is assumed to be negligible, as we are regressing the annual peak flow against the average flow in the low flow season. Therefore, each year we have a pair of data and thus the above assumption is fully justified and in our opinion does not need to be checked (see also Montanari and Brath, 2004; Maranzano and Krzystofowicz, 2004).

*It was unclear to me why was temperature was included in the study. Temperature patterns are discussed in the section on long term persistence, but there isn't an explanation of how temperature relates to the model or goals of the study.*

Temperature as well as rainfall were included to verify whether the hypothesis of the presence of LTP is supported by data evidence in both rivers. We consider both variables to be the major drivers of river flows in both rivers, where both water mass and energy balances determine the response of the catchment. Once we demonstrated that the presence of LTP in the river flow data may be a reasonable working assumption, we use the river flow in the pre-flood season as explanatory variable. We provided in the introduction of the paper a justification for the use of river flow data as a proxy for mean areal rainfall over the catchment.

*The conclusion section is quite short and could benefit from some additional discussion, perhaps something about the utility of the method for other locations and differing catchment sizes. The abstract notes "The proposed technique may allow one to reduce the uncertainty associated to the estimation of flood frequency" – could you could elaborate on this in the conclusion?*

We agree with the reviewer and therefore extended the concluding remarks by discussing the applicability of the method to other cases.

*Minor comments*

*The use of the word "significant" should be clarified (as in the abstract and p 12, line 19). Do you mean statistical significance? At what level?*

We used the term "significant" to mean "noticeable". We changed the terminology to avoid confusion with terms used in statistics.

*Abstract Lines 15-17: I think this would be clearer if re-organized, perhaps: "To exploit the above sensitivity to long term perturbations, a Meta-Gaussian model and a data assimilation approach is implemented for updating the flood frequency distribution a season in advance."* OK

*Abstract Line 20: A word is missing: I would suggest adding "which" before "occurred (ie, "which*

*occurred" or even "occurring" rather than just "occurred")* OK

*P 2 Line 11: an "a" is missing before "long time"* OK

*P 2 Line 15: "associated with" more commonly used than ""associated to"* OK

*P 4 Line 20 Extra word "on" before "the trend"* OK

*P 4 Line 24 I suggest adding a comma after "As stated in the Danube River Basin Management Plan" (since it is a dependent clause)* OK

*P 4 Line 25 significantly rather than significant?* OK

*Page 9 line 18 "was" after a plural sounds strange – perhaps "We applied directional statistics. . ."* OK

*P 10, lines 4 This makes it sound like 0.71 is a sort of cut off. Perhaps "H values above 0.5" would be better here, reminding us that that is the cut-off of interest. Or you could say "H values of 0.71 or higher".* OK

*P 11 line 14 You refer to table 2 but I believe you mean table 3*

Yes, we apologize for the error. It has been corrected in the revised version.

*P 11 line 15 I'm not sure what you mean by "we appreciate" here.*

We changed it to "always found"

*P 12 line 1 Is it really "A goodness-of-fit test" or more an evaluation that model assumptions about the residuals are met?*

We agree that it is an evaluation that the model assumptions about the residuals are met. We revised the text accordingly.

*P 12 line 13 – again, perhaps "we find" rather than "we appreciate"* OK

*P 12 line 19 June is mentioned but not shown on the plots? Was that intentional?*

We regret that there was an error and we meant July. It has been corrected in the revised version.

*P 13 lines 1-3 The text has: "The anomaly in the low correlation coefficient in March previously explained determines an insignificant change in the estimate with respect to the unconditioned distribution." But it appears that the line corresponding to march coincides with the line corresponding to January, not to the unconditioned distribution.*

Yes, we apologize for the error. It has been corrected in the revised version.

*Figure 4 – I find it helpful to add a horizontal line at 0 when assessing homoscedasticity of residuals. Though I'm not sure we need to see these plots. Just describing them in the text is probably sufficient.*

We regret to disagree here with the reviewer as these plots show the evaluation of the behavior of the residuals and so, including them allows us to assume that the linear model proposed can be applied to our data.

*Figures 5-6 Adding a-f markers to each subplot would help with finding the plot being discussed in the text; add labels to y-axes* OK

[revised manuscript text omitted]

---

## Author Response (AR2)

**Real time updating of the flood frequency distribution through data assimilation**

Reply to the editor's comments

*1.        Reviewer #2, Comment starting and ending with: "Also, it would be good to provide more rationale...Readers with a more traditional statistics background..."*
*While I think the authors' addition of other references that use the NQT methods, a more detailed literature review needs to be provided that includes the purpose and flow statistic that NQT has been used previously. I agree with Reviewer #2 that this method is still not widely used and therefore some additional background and context should be supplied. I fully understand the reviewers not adding details of how the NQT method differs from traditional methods but there should be more background in the text. If the use of NQT for flood frequency is a novel use of NQT (which I believe it is) then perhaps it is worth nothing this with associated literature in the introduction.*

Following this comment, we added some detail about the studies already referenced and we added the following paragraphs in the "Methodology section" (page 7, lines 1-3, 8-17):

A meta-Gaussian model (Kelly and Krzysztofowicz, 1997; Montanari and Brath, 2004) is used to model the joint probability distribution between the selected explanatory and dependent variables. The method involves the following steps.

First, the time series $Q_m(t)$, $Q_p(t)$ and $Q_{mf}(t)$ with sample size $n$, where $n$ is the number of years in the observation period, are extracted from the observed datasets. Then, the Normal Quantile Transform (NQT) is applied in order to make their marginal probability distributions Gaussian, therefore obtaining the normalized observations $NQ_m(t)$ and $NQ_p(t)$ and $NQ_{mf}(t)$.

The NQT is a non parametric transformation that can be applied to normalize any arbitrarily distributed random variable. There are numerous applications of the NQT in hydrological studies, to generate flow samples from specified marginal distributions (Moran, 1970; Hosking and Wallis, 1988), to perform Bayesian updating of prior distributions (Kelly and Krzysztofowicz, 1994), to model bivariate distributions with arbitrary marginal distribution (Krzysztofowicz et al., 1994; Aguilar et al., 2016). The NQT is adopted within the Bayesian Forecasting System for river flows (Krzysztofowicz and Kelly, 2000; Krzysztofowicz and Herr, 2001; Krzysztofowicz and Maranzano 2004a, b; Maranzano and Krzysztofowicz 2004). It was also applied for assessing the uncertainty of rainfall-runoff simulations (Montanari and Brath, 2004; Montanari and Grossi, 2008; Bogner et al., 2012), to deseasonalise hydrological time series (Montanari, 2005). Being free of any distributional assumption, the NQT allows one to avoid the selection of a suitable parametric model for the distribution of the considered hydrological variable."

Actually, it can be seen that the NQT has been extensively applied in hydrology. Therefore we believe that it is not worth including its description in the introduction, as the NQT is not a novel contribution of this paper. We would prefer to better describe it in the Methodology section.

*2.        Reviewer #2, Comment starting and ending with: "It was unclear to me...how temperature relates to the model or goals of the study."*
*I did not see any mention of the relevance of temperature to these basins in the introduction. This should be introduced here. The authors state that "we provided...a justification for the use of river flow data as a proxy for mean aerial rainfall..." but this says nothing about the use of temperature.*

*The added sentence in the Methodology section (Section 3) is too late in the text to introduce the use of temperature. If temperature is part of the hypotheses underlying the study, then it should be noted in the Introduction.*

We believe that the estimation of LTP for meteorological variables is a necessary prerequisite in order to better assess the memory properties of the river system. Temperature is important because it is an effective driver of river flows. We better explained the reasons for analyzing temperature in the "Methodology" section of the revised paper, that now reads as follows (page 7).

**"3 Methodology**

In order to address the research question outlined in Section 1, namely, to verify the opportunity of updating the flood frequency distribution a season in advance by exploiting the information provided by the river flow in a given pre-flood season, we perform an analysis of the memory properties of the hydrological cycle in the considered catchments. We first focus on meteorological variables, namely, temperature and mean areal rainfall to check whether a memory pattern is detectable in the weather. Rainfall and temperature are considered as they are the main drivers of river flow, with temperature being particularly influential on the lower values. Then, we turn to the direct analysis of river flows.

We first estimate the Hurst exponent (H) for the considered time series, to verify whether the hypothesis of the presence of LTP is supported by data evidence. Then, we turn to the analysis of the statistical dependence between the peak flow in the flood season and the average flow during the previous season, to empirically check whether updating the flood frequency distribution produces useful results. Results from the latter analysis are assessed in view of the LTP estimation."

Finally, we made a note on the presence of LTP in meteorological data as a possible explanation for the presence of memory between flood flows and the flow regime in the pre-flood season. In particular, we added the following text at page 13, lines 14-18:

"These negative correlations put in evidence that low flows in the winter season may be related to higher flows in the summer season and therefore higher peak flows in the fall season. The latter outcome could be explained by a higher storage during the winter months in the form of increased snowpack, which may be related to the frequency and memory properties of temperature and precipitation data".

*I also urge the authors to review the HESS author guidelines particularly as they pertain to equations, variables, and figures: http://www.hydrology-and-earth-system-sciences.net/for_authors/manuscript_preparation.html. Figure 5, for example, has an unlabeled y-axis. This needs to be fixed. Carefully ensure this manuscript complies with all HESS submission guidelines.*

We apologize for not having fully met the author guidelines. We have carefully revised the mathematical requirements in the manuscript preparation. In this way, we have corrected equations, variables and parameters all through the text (mainly in sections 3 and 4). However, we need to use multi-letter variables in some instances to identify the normalized values of the variables $Q_{\mathrm{m}}$, $Q_{\mathrm{mf}}$ and $Q_{\mathrm{p}}$. Moreover, we cannot use upper-case and lower-case symbols to distinguish between random variables and their realizations as we already use upper-case symbols for our multi-letter variables. Thus, we use the bold character to make a difference between random variables and their outcomes as specified in the manuscript (page 6, line 20).

Regarding the figures, we changed Figures 5 and 6 to include y-labels.

Finally, we would like to acknowledge the editor for her valuable comments and suggestions.

[revised manuscript text omitted]